# Open Access Perceptions, Strategies, and Digital Literacies: A Case Study of a Scholarly-Led Journal

**Noella Edelmann \*** and **Judith Schoßböck \***

Department for E-Governance and Administration, Danube University Krems, A-3500 Krems, Austria
\* Correspondence: noella.edelmann@donau-uni.ac.at (N.E.); judith.schossboeck@donau-uni.ac.at (J.S.)

**Abstract:** Open access (OA) publications play an important role for academia, policy-makers, and practitioners. Universities and research institutions set up OA policies and provide authors different types of support for engaging in OA activities. This paper presents a case study on OA publishing in a scholarly community, drawing on qualitative and quantitative data gained from workshops and a survey. As the authors are the managing editors of the *OA eJournal for eDemocracy and Open Government* (JeDEM), the aim was to collect data and insights on the publication choices of authors interested in OA publishing and other crucial factors such as personal attitudes to publishing, institutional context, and digital literacy in order to improve the journal. In the first phase, two workshops with different stakeholders were held at the Conference for e-Democracy and Open Government (CeDEM) held in Austria and in South Korea in 2016. In the second phase, an online survey was sent to all the users of the e-journal JeDEM in October 2019. From the workshops, key differences regarding OA perception and strategies between the stakeholder groups were derived. Participants strongly perceived OA publishing as a highly individualist matter embedded within a publishing culture emphasizing reputation and rankings. The survey results, however, showed that institutional support differs considerably for authors. Factors such as visibility, reputation, and impact play the biggest role for the motivation to publish OA. The results from both inquiries provide a better understanding of OA publishing attitudes and the relevant digital literacies but also suggest the need to investigate further the enablers or difficulties of scholarship, particularly in a digital context. They clearly point to the potential of regularly addressing the users of the journal as well as communicating with them the more nuanced aspects of OA publishing, non-traditional metrics, or respective digital literacies, in order to reduce misconceptions about OA and to support critical stances.

**Keywords:** open access; open science; open scholarship; journals; policies; publishing strategies; motivation; reputation; ranking; impact factor

## 1. Introduction

The visibility and credibility of a researcher strongly depends on quantitative metrics such as impact factors or citation counts, and there is evidence that open access (OA) publication has a significant impact on the number of citations [1–4]. In academia, gold OA is related to publications made available to the public by authors paying "article processing charges" (APCs) [5–7], but there are also open access journals that do "not charge any fees at all to the author since the costs are covered by institutional subsidies and/or voluntary work" [8] (p. 28), a concept also known as platinum or diamond open access [8], whilst OA repositories and green OA allow authors to self-archive their publications [9,10]. At the same time, altmetrics [11] and dedicated initiatives such as Coalition S [12,13] also play a major role in OA publishing [14]. European funding organizations now have OA policies to ensure that those receiving funding publish their results OA. Initiatives such as the Plan S have been

set as standard by the European Science Foundation [12] and outline the principles by which research funded by grants must be published or made available in repositories or OA journals. There is also an increasing trend toward informal publishing, and characteristics from different disciplinary fields [15], personal factors [16] and digital skills can impact quantitative metrics too.

The question of publishing OA can be seen as evolving from the "if" to the "how." Several recommendations and policies are expected to increase availability and number of views of OA content considerably. In 2016, the EU ministers of science and innovation announced that all European scientific publications should be immediately accessible by 2020 [13]. It is estimated that by 2025, 44% of all journal articles will be available as OA, and 70% of all views will be to OA articles [17]. Several national OA working groups (for instance the Open Access Network Austria) are recommending a transition to 100% OA publishing by 2025 as foreseen by the Berlin Declaration on Open Access to Knowledge in the Sciences and Humanities [18] as well as ensuring that funding is made available for research organizations to cover APCs [19,20]. Finally, Plan S [12] foresees that research funders will mandate that publications generated through grants must be fully and immediately open [13], although currently only few OA journals are compliant with Plan S [21]. In Austria, for example, all academic publishing resulting from public funding must be published online in their final form without time delay (gold OA), and the costs of publication are either to be financed directly by the scientific organizations or authors receive the necessary funds themselves. With the implementation of more OA policies and so called "shock" initiatives aiming for radical change (the "S" in Plan S stands for "shock") [12], personal and institutional factors can be expected to play an important role for publishing cultures, models, and the characteristics of electronic journals in general. As already pointed out by Smecher [22] in 2008, the underlying structures of academic communication are always changing and evolving, involving new methods of peer-review (e.g., "open" review or credits for reviewers), delivery, and subscription. Similarly, the relevance of traditional and non-traditional impact measures, or the necessary knowledge authors and editors are expected to have, are subject to changes. Scholars combine traditional and non-traditional ways of communication, and the dividing lines between formal and informal dissemination of work have crumbled [23].

In the light of the massive global change brought on by COVID-19, the pressure with view to rapid and transparent peer-reviewing and publishing has grown yet again. This means that the argument of OA as being beneficial to society is more convincing than ever, as politicians expect fast publications and data sharing, and society experiences how openness can speed up the impact of science. In an OASPA (Open Access Scholarly Publishing Association) webinar about scholarly communication and publishing in the 21st century, Executive Director of SPARC (Scholarly Publishing and Academic Resources Coalition) Heather Joseph pointed out that the values of scholarly communication have fundamentally changed due to the global epidemic and that there is enormous pressure on policy-makers and funders to support faster and open scholarly communication [24]. While scholars are now demonstrating better ways to communicate, there are also some uncertainties or even conflicting strategies. However, it is estimated that non-OA publications about COVID-19 will grow at a higher rate than that of OA publications [25]. In any case, research institutions are forced to change existing operations, and fundamentally new systems will be needed for new operating realities. New modes of sharing (for instance social media, preprint servers, open infrastructures) and scholarly communication platforms that emphasize community and collaboration [26] will play increasingly important roles. This means that publishing cultures and systems reflecting different values compete with each other, forming a major societal and scientific challenge. Moreover, the role of institutions and ownership of infrastructures should be important subjects of future debate and research. From an individual perspective, these developments do not mean that every author is ready, literate of these technical and value-driven possibilities, or has the necessary support. From the perspective of journal editors and scholarly publishers, there is a need to understand authors' publication choices and to access their perceptions and literacies in this context.

In the research presented here, we focus on OA journals as a dimension of OA publishing. This differentiation is important as publishing formats (journals, books) and their selection as a publishing venue differ strongly between disciplines and research fields [27–29]. This paper aims to shed light on the perspective of members of an OA scholarly community, specifically, the audience of the interdisciplinary OA *Journal for e-Democracy and Open Government* (JeDEM) [30] a journal indexed with Scopus, EBSCO, DOAJ (Directory of Open Access Journals), Google scholar, and the Public Knowledge Project metadata harvester and fully conforming with Plan S [12]. Using a case study with a mixed-methods methodology, the goal of this paper is to investigate the user's views on open scholarship, the factors that motivate OA publishing, and the role of the institutional context and digital literacies. The researchers are the managing editors of the journal, and this investigation is important not only for the continuous development of JeDEM but also to consider the results gained for the research community interested in OA publishing and for other editors who either already publish OA or are moving to OA publishing. The research presented in this paper is based on two inquiries:

(1) Data collected at two workshops conducted with members of this scholarly community, held at two international conferences (CeDEM, Conference for e-Democracy and Open Government, in 2016 in Austria and subsequently in South Korea). The CeDEM conferences draw an audience interested in aspects of the digital, open society and, there, three different stakeholder groups discussed motivational aspects of OA publishing [5–7]. In order to gain more detailed insights from a scholarly community familiar with OA

(2) an online survey was sent to all registered users of the JeDEM in 2019. JeDEM is a unique access point to authors in the sense that it gathers this international scholarly community, a community that is interested in publishing OA. Thus, a diverse target group could be reached.

The results from these studies help to gain better insights into the perspectives held by a specific scholarly community in order to further develop the journal it is associated with but also an understanding of OA publishing cultures, contexts, and the necessary digital literacies authors and editors require, such as the enablers, difficulties, and potentials of open scholarship. This paper thus offers the following novel perspectives and contributions: Firstly, we aim to extend existing studies on OA publishing choices by focusing on an international and interdisciplinary scholarly community and case, hence contributing to research on the socio-political dimensions of OA. Specifically, we highlight authors' perception of key topics and paradigm shifts in OA publishing based on two workshops. Secondly, we present results on authors' evaluation of OA journal publishing based on a subsequent survey of users stemming from a variety of fields and research contexts, with ICT (Information and Communication Technology), governance, and society as a common point of reference. Thirdly, our results can be valuable for academic journals positioning themselves on the digital market, for supporting journal development including author's preferences and digital skills, the role of new standards and research policies such as Plan S criteria (which the JeDEM fulfils), social media integration, and alternative metrics.

## 2. Literature Review

OA is a broad topic, addressed from different points of view and constantly evolving as the impact of digitalization of publication, scholarship, and literacy increases. Given that results from different studies and disciplines inform each other, an interdisciplinary approach needs to be adopted. The focus is therefore on the role of OA in different sectors and research cultures and on how its implementation is coined by institutional policies, personal motivations, and digital literacy.

### 2.1. OA in Academia, the Private and the Public Sector

Open access and open scholarship represent a paradigm shift in several fields. Predicting the future of publication trends and OA is crucial for different stakeholders such as publishers, libraries, and funders [17]. With a view to the different types of OA and OA publishing formats, many journals

offer so-called green OA, which is the free access to publications via self-archiving by the authors [9]. While this is a convenient method for keeping the cost regarding publications low, it impacts the quality of the available versions. This is due to the fact that authors are not always allowed to provide the final publication, but pre-versions, which could be incomplete or erroneous in comparison to the final paper. In addition, publication venues may offer a golden OA approach, where authors are required to pay APCs to make their publication free to readers [10]. Gold, diamond/platinum, and green OA differ mainly in their support costs and in the roles they can play in the scholarly communication [10]. The golden approach provides the public with free access but still imposes publication fees (article processing charges, APCs) per article on its authors, although only 28% of the journals charge APCs [31]. Several funding options are available, and whilst these fees may be covered by the authors' institution, they can be a problem for smaller institutes, independent researchers, or institutes in developing countries.

Arndt and Frick [32] undertake a case report and stakeholder analysis of motivational factors for the purpose of OA training strategies of librarians as useful starting points for marketing. Five key areas are derived: the political environment, the work environment, the professional environment, organizational environment, and projects in the environment, with scientists being classified under work environment. Within the scholarly community under investigation, authors publishing OA come from different fields or stakeholder groups: academia, the private and the public sector [33] as well as from several other industries: libraries, publishers, and scholarly societies [17].

In academia, OA is foremost related to publications and their availability to the public without fees of any kind. On the one hand, there is strong evidence that the factor of a publication being OA has significant impact on its citation numbers: visibility and credibility of a scientist strongly depend on quantitative metrics such as impact factors or citation counts [1–3]. On the other hand, other factors such as the broader reputation of a journal, the quality of content, and publishing cultures should not be ignored.

In the private sector, OA is one key component in the paradigm shift from closed research and development environments toward open innovation principles and can refer to the authors and readers of OA articles [34]. The idea behind this approach is to open up individual development processes and share data and knowledge with others. By doing so, narrow-minded, "closed" environments can be opened and innovation and product cycles can be accelerated by introducing new ideas, concepts, and best practices as well as "novel ways to create value" [35] (p. 41). However, not every involved party only sees positive aspects in these changes. Some voices claim that there may be pitfalls associated with diffusing relevant knowledge [36] and disclosing company essentials [37].

Beyond the two aforementioned fields, another area that it is worthwhile studying is the public sector. The global shift toward giving free, open, online access to the results of publicly funded research has been a core strategy of the European Commission to improve knowledge circulation and thus innovation. Openness has been supported by official EU directives for several years, for example, by the Public Sector Initiative (PSI) Directive [38] for public sector information platforms [39,40], but also more specifically by the general principle of OA to scientific publications in Horizon 2020 and the Open Research Data Pilot [39,41]. Due to EU-enforced regulations regarding the necessity to publish public sector information in order to foster innovative business ideas and EU-based economy as a whole, several issues arise such as data privacy, conflicts of interests, and challenges regarding data publishing and data curation.

## 2.2. Research Cultures, Attitudes, and Personal Strategies

Research cultures and disciplines play a role in motivational factors for OA publishing. Based on two online surveys conducted with European authors from 2009 to 2011, by Fry et al. [42], that compared the perceptions, motivations, and behavior of authors according to their discipline, they explored the cultural characteristics of OA-friendly research cultures such as physics, economics, and clinical medicine. Their results show that cultural characteristics influence perceptions, motivations,

and behaviors toward green OA. This study points out the need to consider authors' motivations and the role of disciplines when studying OA. The authors also note a slow adoption rate of OA outside those disciplines that already have a tradition of publishing with reputable open access journals, e.g., authors from science and medicine disciplines who publish with PLOS (Public Library of Science) journals. However, this study was undertaken before the active support of OA policies by institutions became more established, which warrants further investigations into the role of disciplines in OA. Disciplines also seem to play a role regarding the perception of the quality of OA publications. Based on a Spanish survey on researchers' attitudes toward OA publishing, there does not seem to be a clear position regarding quality of OA journals among researchers, even though the vast majority of those surveyed (76%) believe OA is beneficial and have published OA in the last 5 years (70%) [43]. However, asked whether OA leads to an increase of publication of poor-quality research, 31.2% agreed or fully agreed with this perception, which basically means that researchers see OA articles as synonymous with research of low quality. There were some differences between disciplines, for instance between social sciences (65.8% disagreeing) and education (50% disagreeing) or psychology (50% agreeing), chemistry (40% agreeing), and agriculture (39.1% agreeing) on the either side of the spectrum [43] (p. 726). This points to the potential persistence of OA clichés and misperceptions that likely differ between disciplines and research cultures.

Previous research defined some broad categories of authors' motivational factors for OA publishing. Samrgandi [16] undertook an investigation based on dualistic motivation theory and dissertations, examining authors' awareness of publishing OA as well as the key factors that motivate authors to do so. Intrinsic and extrinsic factors were examined, with the following key factors emerging from this analysis: Authors believed in a richer data repository (extrinsic factor) or benefits to authors' careers or prestige or reputation (intrinsic factors) [16]. While the most frequently identified motivation was extrinsic, the other most significant factors were all intrinsic.

Several studies look at scholars' attitudes regarding publishing. Shehata, et al. [15] undertook that effort using a grounded theory approach with 40 researchers in four universities and focused on informal publishing and digital trends. They concluded that there is an increasing trend among researchers toward informal publishing and dissemination throughout the scholarly communication cycle. Three types of scholars are defined: conventional, modern, and liberal. Each of these types of scholars carries different beliefs regarding the scholarly communication process, and authors notice the trend toward using informal channels for publishing or scholarly discussions [15]. These results can present a starting point for further research focusing on the adoption of different formal or informal strategies of OA publishing within research cultures or communities. A few studies investigate how authors select a publication venue or repository or the factors influencing publication choice in OA [27–29]. While there are not too many longitudinal studies, addressing this issue has become increasingly popular, with researchers focusing on specific cohorts such as early career researchers or the faculty of a specific university. Several research areas tie in with the aspect of publication choice, for instance, the role of institutional context and policies, personal attitudes, and digital literacy. Repeated surveys have been conducted, for example, by publishers such as Springer by directly contacting registered users in order to access author perceptions and preferences through their registered users [44,45]. Taylor and Francis also conducted a researcher survey in 2019 aimed to gather their views, around publishing, publication venues, and the future of scholarly communications [46]. Apart from these researcher surveys by publishing companies, several studies have been undertaken at the European Union level, for instance about scientific information in the digital age, pointing toward a general positive attitude toward OA held by different stakeholders (including governments), with green or gold OA identified as the preferred ways that public research policy should facilitate [47]. Important input from the future of OA policy also comes from a survey on FP7 (European Commission's FP7 funding programme) project coordinators. Strikingly, for 60% of respondents, understanding issues around copyright and licenses to publish is difficult or very difficult, pointing toward the need of communicating OA skills, as well as reimbursement options [48]. The OA Survey Results of the

EUA (European University Association) give an overview about awareness levels and the institutional context of leaders, librarians, and researchers, specifically the evolution of OA policies at the European level. At the time of the survey, 60% of institutions had OA in place [49].

Several studies have been undertaken to identify emerging trends in scholarly publishing, for instance, by focusing on authorship practices of early career researchers and their take-on of digital innovations. Nicholas et al. identify two factors that hinder the adoption of new practices: constraint by conventions and precarious employment environment, pointing toward specific hurdles of this target group and underscoring the strategic importance of high-impact factors [50], particularly in situations where career pressure might influence attitudes [51]. More studies are needed to further investigate this issue for different target groups and how to better align OA perceptions (which are generally positive [43]) and the actual publishing practice. OA perceptions and motivational factors have also been addressed by Kim et al. studying institutional repositories and faculty adoption, who point out that 80% of their surveyed faculty members agreed with OA but often had no awareness of repository offers [52]. Next to cost and benefit factors, they identify two other factors hindering such activities: contextual factors and individual traits including technical skills. Moreover, altruistic motivations are important for taking action in this context [53]. A wide bibliographical review of OA perceptions in an institutional context of one university was conducted by Serrano-Vicente et al.: their study shows significant differences in opinions regarding OA journals, again pointing toward the gap between general opinion and actual practice and the importance of academic recognition for making publications accessible [54]. Motivations for selecting a journal have been studied at four large North American universities. Interestingly, OA was the least important attribute in this study and the general public were the least important audience [55]. Such results ask for an extension of studies in scholarly communities that might be more focused on OA publishing or by including more stakeholders, in order to better derive success factors and hurdles of the broad target group for OA publishing.

Digital literacy plays an important role in academic communication as authors increasingly use digital channels and tools to support and track the dissemination of their published work and expect journals and their community to do this too. Digital literacy is understood as the ability to use information and communication technologies to find, evaluate, create, and communicate information [56] and is seen as a set of interdisciplinary competences central to society. Digital competences include a range of skills, from the basic skills required for using digital devices and everyday life, to those required for work, creative expression, information processing, learning, and taking informed decisions [57]. Authors, readers, publishers, and other users therefore must develop digital literacy, information competences, and practical approaches [58] in order for them to know how OA influences their publishing strategies. Furthermore, with the increasing reliance on metrics in research [59] and as discourse concerning metric manipulating intensifies [60], digital literacies play an important role for influencing publication metrics and also for identifying misconduct or misuse within publishing systems. One could further postulate that a lack of digital literacies could even hinder critical stances on the publishing system or activism toward open science, understood for instance as the development of competent views on the manipulation of metrics or databases. Lack of a digital skillset could also make it harder to identify the unwanted effects or inappropriate applications of metrics [61].

Whilst the literature focuses on OA in specific contexts, such as the authors' attitudes, or their disciplines, or the institutional policy, only a little research looks at the cross-section between the different aspects. We suggest that there is a clear need to gain new insights from scholars who are already publishing in OA journals, in order to investigate their knowledge about the role of institutional context and their personal perceptions and strategies. We suggest that institutional policy, individual strategies, and scholarly literacies, and, increasingly, digital literacies play a role for publication choices. This case study therefore focuses on two main research questions: first, what arguments for open scholarship do different OA stakeholders have, and second, how do authors of an OA scholarly community choose a publication venue?

## 3. Methods

The case study approach was used as it allows the investigation of a phenomenon within its real-life context and the use of multiple sources of evidence whilst retaining a real-world context and perspective [62]. According to Yin, it is a particularly suitable method when the focus is on a contemporary event and its context as well as the various interactive processes [63] and the dynamics within this context [64].

The research design used to answer the questions in this case study is a mixed-method design. We use a qualitative approach to study the results gained from two workshops [6,7], and a quantitative and qualitative approach to study the results from an online survey based on closed and open questions. This paper combines the results gained from stakeholder workshops and a re-analysis of data produced for operational purposes (survey results) for our specific journal and interdisciplinary scholarly community. Workshops were chosen to address the first research questions. A qualitative approach was selected as qualitative research is "suited to promoting a deep understanding of a social setting as viewed from the perspective of the research participants" [65] (p. 38). While there is a little disconnect in time between the two phases of the methodology (see Section 3.1 Research Design), the focus of the first part on stakeholders' views on open scholarship allows us to pinpoint the perceptions of different stakeholder groups for the selected case study (see Section 3.2 Case Selected). The emphasis here is on exploration and description. Quantitative research is used to investigate relationships to investigate relationships and weight relevant factors. By combining both research methods and triangulating the result, the researchers are able to address a complex and multi-layered phenomenon, and results gained from one method inform the other [65].

### 3.1. Research Design

This mixed-methods case study focuses on the following research questions:

RQ1: What arguments of open scholarship can be found by different stakeholders?

- What are the enablers and barriers of OA?
- Where do they differ in their arguments?

RQ2: How do authors choose a publication venue?

- What is the role of publishing cultures, institutional context, policies, and support?
- What is the role of digital literacy?

The research, as shown in Figure 1, is based on a mixed-methods design.

To answer the first research question, researchers held two workshops at the international CeDEM conferences (Conference for eDemocracy and Open Government held 2016 in Krems, Austria [7] and in Dague, South Korea [6]). The workshops aimed to collect data on three different stakeholder groups' opinions and views about the motivational or strategic aspects of OA publishing. Building on these results, an online survey was developed and distributed to users of JeDEM (October 2019) with qualitative and quantitative items that addressed several aspects in more detail, such as OA policies and institutional support, personal strategies, the perception of specific journal features (e.g., metrics), and publishing quality [33]. Users of JeDEM and the participants of the CeDEM conference community represent different stakeholder groups but with an interest in both the journal and the conference. The researchers organized the workshops and, as managing editors of JeDEM, were able to give the different communities and the journal access to the user database.

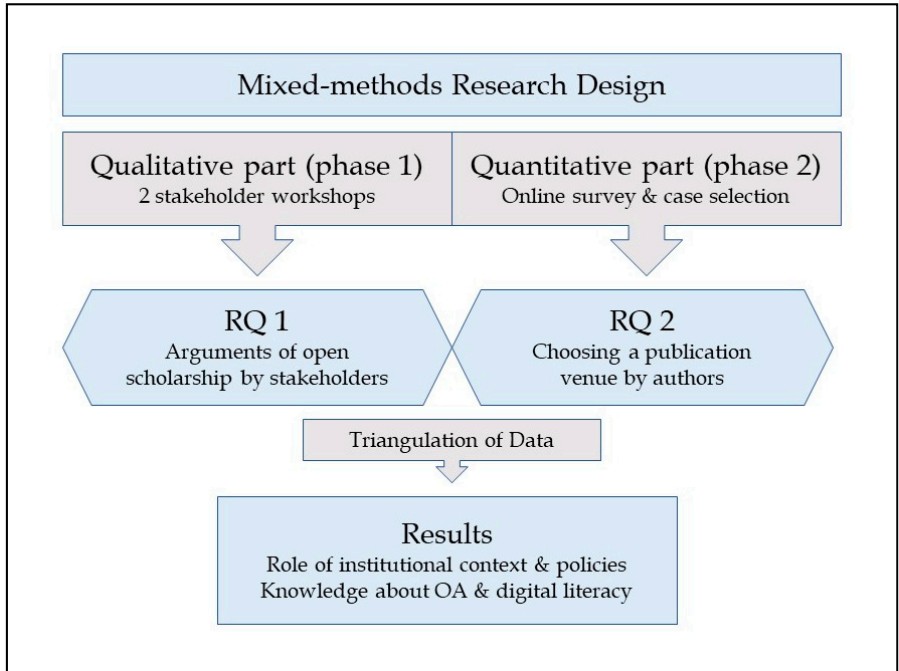

**Figure 1.** Research Design.

### 3.2. Case Selected: OA Journal of E-Democracy and Open Government

JeDEM, the OA *Journal of E-democracy and Open Government*, was selected as one of the two authors set up the journal in 2009 and both have been the editing managers since the beginning. The first issue was published in 2009, so it is a relatively young OA e-journal with a wide range of subjects and research fields, and a call-for-papers and articles such as those on e-democracy, e-participation, open government, and open data in the public sector, which cover diverse topics. Hence publishing with JeDEM attracts a wide range of authors and readers from different disciplines, communities, and sectors. The added value of our case furthermore lies in its interdisciplinary target group that comprises different universities and research cultures. It includes researchers interested in OA publishing from more governmental and practical contexts next to classic research institutions, covering different faculty levels. It is also worth pointing out that the journal's scope focuses on topics relevant to the digital society, so findings related to digital literacy have to be interpreted in this context.

Using the classification by Edgar and Willinsky [66], the journal can be described as an independent interdisciplinary scholarly journal sponsored and published by an academic department (Department for E-Governance and Administration) that focuses on the journal topics and is itself interdisciplinary. JeDEM follows the platinum/diamond OA road [8], and is indexed with EBSCO [67], DOAJ [68], Scopus [69], Google Scholar [70], Crossref [71], the Public Knowledge Project metadata harvester [72], and SCImago [73].

JeDEM is available on an online platform only, the Open Journal Systems (OJS) developed by the Public Knowledge Project [72]. By the end of 2019, JeDEM had 798 registered users, and 178 articles had been published. The editors use double-blind peer review, and the journal has a 42% acceptance rate. The staff are all faculty members, with costs only for technical support, proofreaders, and designers. JeDEM offers the DOI (Digital Object Identifier) system as a persistent identifier for the articles published, and, in terms of impact, has a journal-h-index of 22 (2019, based on Harzing's Publish or Perish program, source: Google Scholar), 156.9 cites per year, and 10.7 cites per paper/year (2009–2019). By 2019, the journal was fully compatible with all the Plan S requirements [12]. Publications in JeDEM are not hybrid and are available immediately with a Creative Commons Attribution license (CC-BY), which is also in line with the Plan S standard [12]. The public dashboard of JeDEM and its impact are

available on SCImago, which has been available from June 2020 and will be complete in 2022 once it has received data from Scopus for 3 years [73].

*3.3. Workshops*

The goal of the two workshops was to gain qualitative data on OA authors' publishing strategies, their motivations for publishing OA, and the role of institutional OA policies in order to derive common themes and key differences. The workshops addressed participants' opinions and strategies regarding OA publishing in various application domains found in academia, industry, and policy-making. Data were collected about personal experiences, in particular the role of institutions, and the driving motivation behind following the OA paradigm.

Workshop 1 and workshop 2 both consisted of two parts. One of the editing managers (who are at the same time, scholars from the Department for e-Governance and Administration) started out with a short introduction to the overall topic, an overview of OA publishing (including the example of JeDEM.org), and an overview on OA policies and funding policies (e.g., the OA policy of Danube University), in order to set a common knowledge frame for the discussion. Whilst both workshops were run in a similar manner, the first workshop also included the OA and publishing policies of Danube University, due to the conference location and the availability of the officer responsible for the university's OA policy. In workshop 1, participants were split into groups reflecting the different stakeholder groups; academia, industry, and policies. Participants self-selected their relevant groups. Participants in workshop 1 were presented with a pool of initial questions regarding OA policies and experiences and were asked to crowd-select questions of importance and, afterward, to answer these crowd-selected questions. Each group discussed 2–3 questions and, afterward, discussed the results in the plenum. Workshop 2 followed a similar structure, but because there were fewer participants in workshop 2, discussions were held in plenum throughout the whole workshop. In total, there were 23 participants at the workshops and 4–7 per stakeholder group.

*3.4. Online Survey*

In order to gain further insights from the target group, in October 2019, an online survey was sent to all registered JeDEM users (798 registered users). Qualitative and quantitative items addressed aspects such as OA policies and institutional support, personal strategies, the perception of specific JeDEM features, and its publishing quality. Additional questions were asked in relation to alternative publishing metrics such as altmetrics or knowledge about specific OA policies. The Danube University Online Survey Tool ("EvaSys") was used for the design of the survey, the questions were developed on the basis of the literature review above. The questions were split into thematic groups and considered the following dimensions:

1. Demographic data
2. Strengths and weakness of academic publishing and open access
3. Reasons for publishing or not publishing with JeDEM
4. Institutional support for publishing OA

The first announcement made on the JeDEM platform was made on 24 September 2019, followed by a reminder as an announcement made on the platform and an email to all the registered users on 14 October 2019. The online survey was closed on 25 October 2019.

**4. Results**

The results of the workshops were originally used to develop the journal according to the users' perspectives on OA and to understand their needs. The need to continuously adapt the journal's policy to the new OJS platform updates, new data protection regulations (e.g., GDPR), digital publishing and research standards (e.g., Plan S), and modes of publishing (e.g., altmetrics), and the university's OA policy and support meant that further insights into the user community were necessary. Thus,

the survey was set up to provide new insights into the community's views on academic publishing, publishing with JeDEM, and their perceived institutional support for OA.

*4.1. Workshop Results*

In general, the results from the workshops held at CeDEM16 [7] and CeDEM Asia 2016 [6] confirmed the literature [17,32,34] about different stakeholders having different views on OA publishing. Particularly researchers or people within institutions and funding organizations have very different attitudes and approaches about OA publishing. Qualitative data collected from the three specified stakeholder groups within this scholarly community: (1) practitioners, (2) policy-makers, and (3) academics show that they discussed several topics, such as the role of publishing cultures and institutions for personal strategies, and the summary in Table 1 shows not only differences between the groups but some overlap too.

Whilst several key differences regarding the assessment of publishing strategies and OA between the stakeholder groups could be derived, both practitioners and academics saw OA as a way to increase visibility of their paper, highlighting journal ranking and impact factors as important criteria. While academics focused on reputation as the main criteria, they were also eager to spread their ideas, so reputation was partly contrasted to open scholarship. As a tendency, policy-makers took on a rather critical stance regarding the impact factor, noting that "some papers are always cited" (participant from [7]), even if they are not good. Some policy-makers also noted that there was no evidence for added value in developing OA for the benefit of progress of knowledge in the scientific community.

Participants strongly reflected on OA publishing as a highly individualist matter that is embedded within a publishing culture that strongly emphasizes reputation and rankings. With a view to general publishing cultures and expectations, researchers noted that working for a journal (e.g., editorship, reviewing) is also associated primarily with reputation (particularly since most review work is done for free). Established channels and non-OA journals are preferred, as they are better known within the researcher community, although participants coming from academia did not relate this to better quality ("they are simply better known"). In that sense, OA publishing culture was somehow contrasted to well-known channels and prestigious journals. This opinion can be expected to change given the high level of institutional support for various forms of OA these days. Academics also noted a common stereotype associated with OA: free of charge is associated with lower quality in the context of their research communities. This is in contrast to the opinion of policy-makers, who do not attach as much importance to impact factors (and in contrast to practitioners, who, as a tendency, also emphasize journal rankings).

There was a consensus in the data from the three groups that OA strategies differ considerably within the research fields, and that OA free of charge is seen as a good opportunity for students. Academics emphasize that there is a lack of support for this particular group of authors. This group also noted that even if costs are not an issue, the motivation to publish OA is small, albeit the important role of metrics such as Google Scholar was emphasized. This could be interpreted as lack of intrinsic motivation [16] for publishing OA (potentially read as a lack of inclusion of OA in career plans) and certainly is an argument for official policies and support in favor of OA, in particular the support of OA versions of journals with high impact factor. At the same time, offering education about the role of OA, metric systems, and digital literacy could help with increasing both intrinsic and extrinsic motivation amongst academics.

Regarding institutional policies, policy-makers were aware of the requirements of OA publishing for EU-funded projects, and they were lacking measures to increase readability of texts. For both practitioners, the institutions' policy on publishing is crucial. Furthermore, academics were well aware of their institutional policies, with some noticing restrictions. They also noted that U.S. policies and models play a strong role for Asian publishing strategies. Some radical OA activism, e.g., the insistence on OA publishing by authors, was mentioned as opportunity: however, this was seen in potential contrast to career management.

**Table 1.** Paraphrased results, stakeholder stances according to open access (OA) dimensions (workshops 1 and 2).

| Dimensions | Practitioners | Policy-Makers | Academics |
|---|---|---|---|
| Aim | Getting the most out of a paper. | | Getting the most out of a paper. |
| Quality | Importance of journal rankings. Tension between quality and reputation. | No evidence of added value in developing OA for the benefit of progress of knowledge in the scientific community. Unknown correlation of quality/numbers of OA publications and scientific progress (improvement within research field). Unknown correlation between quantity and quality in the sense of developing a domain. Criticism of impact factor (some low-quality papers are often cited). | Importance of ranking, reputation, and impact factors. Tension between importance of reputation and distribution of work. Personal resources are also tied to reputation of a journal (reviewing is done for free). Reputation of other researchers and their publishing venues as important factors. Better visibility of established journals (non-OA journals are not better quality but better known). Free publishing is associated with lower quality. |
| Costs | Advantage of OA publishing for students due to crucial role of fees. | | OA as market issue: Google Scholar plays a vital role. Lacking support system for students. OA as an attractive option for no-names, students, or those with no money or organizational/university support. Focus on publishing in high-value journals and conferences: Why publish OA if costs are no issue? |
| Disciplines | Differences between research fields. | | Differences between research fields. |
| Policy | Institutional policy on publishing is crucial. | This is an issue of higher education institutions. EU/EU-funded projects require OA publications. Regulations for publication do not provide much flexibility. | Organizations and institutional policies demand publishing venues. Flexibility: in some cases there are restrictions but not in all. Asia follows the U.S. model. |
| Role | Lack of support for students (professors do not normally publish with students). | | Role (staff or student) is crucial regarding support. |
| Other | | Wishes for measures for more readability of articles. | OA activism: radical or destruction of career? |

## 4.2. Survey Results

The results of the workshops were used with the aim of developing JeDEM according to the new standards such as Plan S, and to adapt to the new dimensions of the new OJS 3.1.2.0 platform. On the basis of the issues raised in the workshop, a questionnaire was developed in order to see whether the results are still valid. In October 2019, the JeDEM journal platform was updated to OJS 3.1.2.0 and announced together with the online survey. This online survey was developed to gain users' insights toward academic and open access publishing, open access, institutional support, the journal, and the new features offered by the new JeDEM platform. The questions focused on the demographics, academic publishing, publishing/not publishing with the JeDEM, and institutional support for OA.

In all, 57 surveys were completed, which indicates a 3.3% return rate. Although this is a low rate of return, by triangulating the answers to the quantitative and qualitative questions, it is possible to gain some insights about the JeDEM users ("demographic data"), how they view academic publishing and OA publications ("strengths and weakness of academic publishing and open access"), the journal ("reasons for publishing or not publishing with JeDEM"), reasons for publishing (or not) with OA journals and institutional OA support ("institutional Support for publishing OA"). The questions were split into thematic groups and considered along the following dimensions:

1. Demographic data
2. Strengths and weakness of academic publishing and OA
3. Reasons for publishing or not publishing with JeDEM
4. Institutional support for publishing OA.

### 4.2.1. Quantitative Measures

Demographic Data

Of the 57 surveys, 72% were completed by men 28% by women. Overall, 64% of the responders were from the 35–54 age bracket, 59% came from higher education, 24.6% from the public sector, 15.8% from an non-governmental-organisation (NGO) and 10.5% from the business sector. Moreover, 47.4% stated that they were researchers, and in terms of education, 74% had a Ph.D. or higher degree.

Strengths and Weakness of Academic Publishing and OA

The users pointed out that their 3 most important reasons for publishing OA are that it allows for a high availability and visibility for the paper (77.2%), because of the journal's reputation (38.6%), and due to high citation impact (35.1%), closely followed by the importance of global access to knowledge (33.3%) (see Figure 2).

**Figure 2.** Important factors for publishing OA.

In terms of quality, 28.3% of the users rate the quality of OA as very good, 43.5% as good, and 28.3% as fair. Further, 67.3% had not heard of Plan S, 12.2% had heard of Plan S but were not aware of what it

implies for their work, and 20.4% were aware of it and knew what it means for their work. A similar picture emerged from the users' knowledge about altmetrics: 59.2% did not know what altmetrics are, 18.4% have heard about it but were unaware of what it means for their work, and 22.4% knew what it implies for their work. Whilst some of the users saw altmetrics as very important (11.1%) or important (33.3%), the majority (55.6%) did not see it as important. Nonetheless, the majority believe that they know enough about OA publishing to make proper decisions regarding academic publication (72%).

Reasons for Publishing or not Publishing with JeDEM

According to the survey, 40.4% had heard about JeDEM through their colleagues or network, 40.4 through an online search, and 33.3% had read an article that had been published there. Furthermore, 66.7% of them had submitted a paper to the journal.

The results from the registered users show that the majority agree or strongly agree (89.8%) that JeDEM publishes high-quality articles, and 84.4% believe that JeDEM has a good reputation. For most (95.9%), the factor that JeDEM is an e-journal plays an important role for their publication choice, and for 89.6%, the idea that JeDEM is an open access journal is a reason to publish in it (Figure 3). Electronic publishing, high quality, open access, and reputation thus all play very important roles for publishing in JeDEM, although for our user group, the e-factor and quality are slightly more important based on this first user survey.

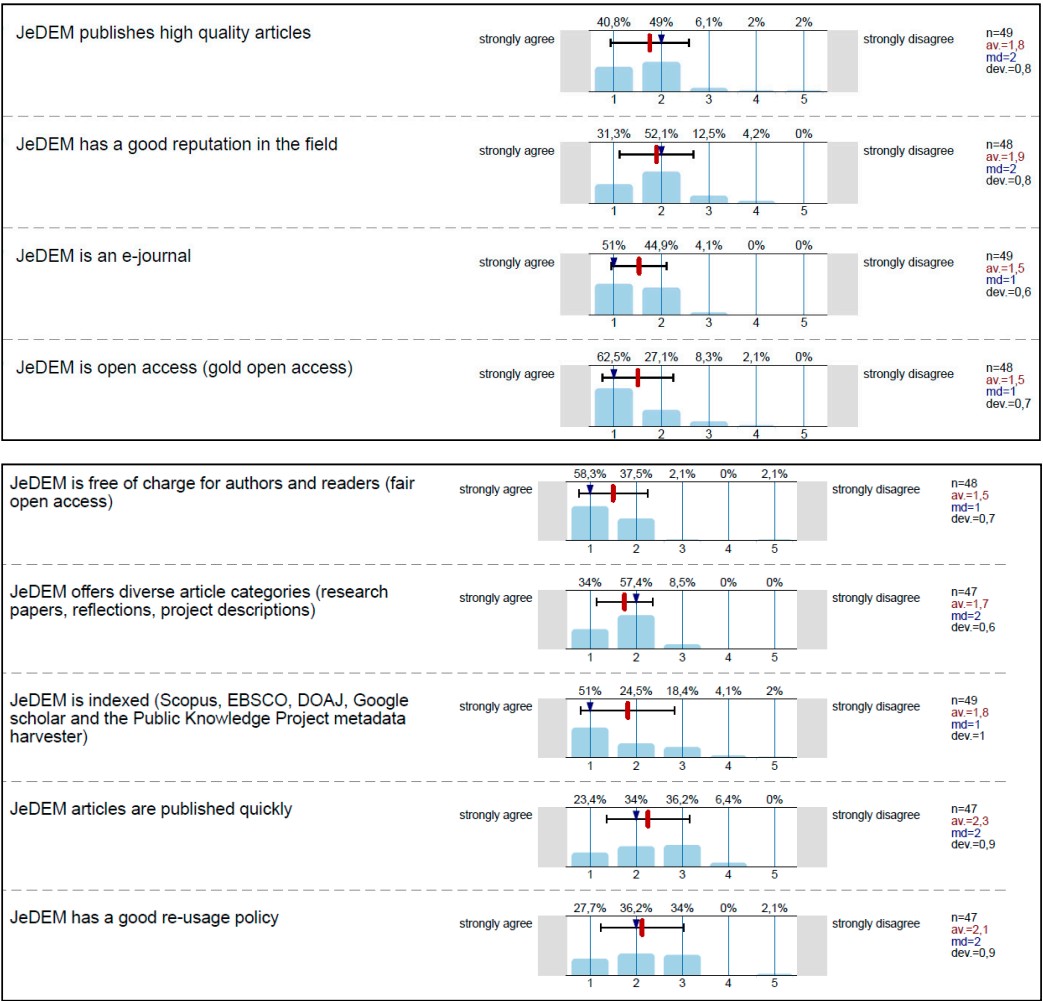

**Figure 3.** Results about why authors chose to publish with OA *eJournal for eDemocracy and Open Government* (JeDEM).

Although the journal is OA and does not demand article processing charges (APCs), only 58.3% strongly agreed that neither authors nor readers need to pay, and only 51% knew that the journal is indexed (with Scopus, EBSCO, DOAJ, Google scholar, and the Public Knowledge Project metadata harvester). Although all articles are published with a CC-BY license, 36% did not think that the journal had a good re-usage policy. As previous research has pointed toward the difficulty of understanding issues around licenses [43], further investigations of this more detailed aspect of OA skills and digital literacy would be fruitful.

Institutional Support for publishing OA

A majority of users pointed out that they had no institutional support for OA (31.6%). Overall, 22.8% selected that their institution had an OA strategy and/or their institution supported publication in gold and fair OA journals (29.8%), where fair is understood in terms of the Fair Open Access Alliance (FAIR [74]) as shown in Figure 4.

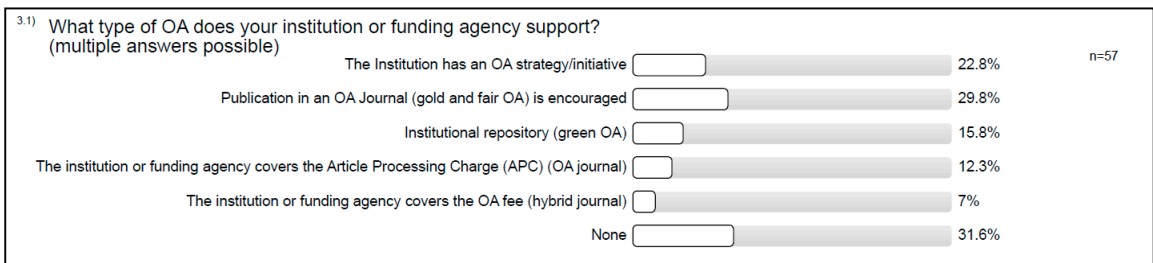

**Figure 4.** Type of institutional OA support.

### 4.2.2. Qualitative Measures

The qualitative items of the survey focused on the same dimensions as above, excluding the demographic data.

Strengths and Weakness of Academic Publishing and OA

Regarding authors' view on the academic publishing landscape of today, we asked an open question about what authors liked and did not like about publishing academic work in general. Here, authors emphasized personal and professional development (foremost the understanding of phenomena and fostering of debate, inspiration, and staying up to date), and aspects of community and sharing (co-creation of knowledge and sharing with the research community and also with the public):

*"distribute research and new ideas internationally; receive recognition in the field"*

*"I can contribute to the public and academic debate around issues that today plague democracy and its governance institutions."*

*"It allows me to share in-depth research with a broader audience, seeking to advance ways of thinking on the topic."*

In the area of recognition and reputation, citations or credits were mentioned most often, hinting at the fact that personal accomplishment seems to be of higher value than general visibility:

*"It can be accessed by those interested in it, and/or who work in fields where it can make an impact."*

*"It contributes to the existing body of knowledge and engages with the scholarly community with different schools of thoughts, hence a great learning opportunity. Also, it increases visibility and high citation."*

*"It helps you to go higher in rank also it helps to extend knowledge to many people."*

Contributing to the public and practical impact was as important as contributing to the professional debate—something that could be seen as a characteristic of the JeDEM user group:

*"Publishing my work gives me moral standing to discuss what I believe in and also share my knowledge with the people whom I can not reach thus contributing to the world of academics."*

*"Provides an opportunity to share and promote my research, and, more importantly, for this research to have some wider professional and societal impact."*

Another mentioned that motivations were related to the quality of published work such as rigor and complexity, mix of theory and practice, and fast access.

Regarding aspects that are less motivating for authors, the publishing process was the category most often mentioned, before the nature of publications and publishing scene in general. Strikingly, authors mainly criticized long review processes and publication procedures, e.g.,

*"(commonly) insanely long publication procedures, with multiple delays—and a corresponding massive delay of any public resonance"*

*"too long for the majority of journals and special books"*

*"reviews take sometimes way to long, up to 1 year for first decision (personal experience). This is unacceptable, especially for Ph.D. students that fight the clock."*

Authors also clearly dislike their own workload and demands in relation to publishing as much as unthorough reviews and rigid processes (when it comes to digital publishing). Regarding OA, this criticism played a subordinated role compared to the publication process in general; however, closed communities and criticism of paywalls and lack of public accessibility were among the most mentioned. This would undermine "the value that research, particularly publicly funded research, can deliver to society."

Reasons for Publishing or not Publishing with JeDEM

Regarding additional reasons to publish in JeDEM, authors emphasized that they are drawn to the journal because of its affinity to the research area they are working in and its OA setup. Thus, our target group recognized JeDEM as a niche journal and consider OA as a major factor for this decision. Furthermore, the high academic standard from a stable publisher is mentioned.

Regarding reasons not to publish, a lack of indexing (*Web of Science*) and visibility was mentioned next to bad review experiences or time issues. Some authors perceived the articles to be of mixed quality and some pointed toward country-specific restrictions that would not allow authors to publish freely where they want:

*"The indexing of the journal is very important, particularly the acquisition of the Impact Factor (or a path toward one) and indexing with CORE. In [my country], researchers are not really free to publish where they want."*

Similar to thematic compatibility for publishing, most negative points for not publishing in JeDEM related to authors working in a different research area. For the governance context, one author mentioned that he or she is not in favor of publishing in journals with a CC-BY Creative Commons license, which could point toward a clash from a practitioner's side, but also highlights a distinct publication strategy.

Institutional Support for publishing OA

The results show that institutions want their staff to publish their articles in journals ranked by SCImago or those that attract higher assessment points. The users of JeDEM pointed out several times that there is no or only limited institutional funding for OA journals:

> *"There is general acceptance but no actual support in terms of tangible funding."*

On the other hand, some users pointed out that the institution may accept OA and support the use of scholarly blogs or the use of open datasets as well as collaborations with other institutions in order to be able to access OA journals:

> *"Open research approaches such as scholarly blogs and open datasets."*

Finally, some users where not sure whether their institutional policy supported OA or not. In line with previous research pointing toward a lack of awareness of OA opportunities [48], we stress the importance of studying the adoption of new publishing offers and the factors that might hinder it.

## 5. Discussion

The aim of this research was to consider the arguments of open scholarship held by different stakeholders, to investigate how authors choose a publication venue, and to assess their knowledge on different OA aspects. The case study was aimed at delineating arguments on the basis of different user groups, to understand authors' view on academic publishing culture and OA publishing, the role of institutional context, and their knowledge and literacy concerning OA and metrics. Two workshops that were to inform the future development of the OA journal JeDEM preceded a survey to further investigate the motivation for publishing in an OA journal by considering the authors' personal attitudes and institutional support. The results from both inquiries point toward the implications of different OA contexts and literacies and different enablers or difficulties of open scholarship.

The results further demonstrate a potential for communicating details of OA initiatives and non-traditional metrics and are meant to inform subsequent studies that look at how authors' perceptions and strategies differ through different locations or research cultures. We would particularly encourage comparative studies investigating different scholarly communities and user groups in addition to the quantitative surveys that have been undertaken so far for different cohorts (such as early career researchers or universities) or by publishing companies. This can lead to a better understanding of OA publishing cultures, discourses, and norms and help to investigate authors' and institutions' needs for a more nuanced understanding and more empowerment. Previous research pointed to conventions and employment environment as factors that hinder the adoption of new OA practices [47]. As it seems that impact factors and pressure on careers [45–47] and from work environments are likely to persist in academia, scholarly-led OA publishing can benefit from acknowledging this situation, while communicating new practices, digital literacies, and support from institutional policies. Our survey results showed that authors do not always know about such policies or how more specific ones might impact their work. While there is a trend toward using informal channels for publishing and gaining visibility [15], researchers will also benefit from enhancing their digital literacy regarding more nuanced aspects of OA publishing. Our study also confirmed the potential for more official strategic support, with only 22.8% identifying an OA strategy of their institution.

The case study approach was seen as particularly useful for this type of research, as the results gained can help contribute to and develop new theories about the topic investigated [75] the results highlight two findings: First, it supports the results of previous studies that journal reputation and citation impact are among the most significant factors in OA publishing, together with visibility and accessibility. Previous research has framed those as intrinsic factors [16], related to an individual's career planning. It might be that OA publishing is not sufficiently recognized in researcher's career plans, and this asks for further investigation. Second, most authors do not mention institutional support as an important factor. This could point toward a missed potential for institutional policies to support

intrinsic motivations, or even stronger and more idealistic, intrinsic motivations as a characteristic of our scholarly community. We therefore suggest that researchers needs to develop a more nuanced understanding of intrinsic factors of OA publishing, particularly as reputation could have both intrinsic or extrinsic qualities. We suggest considering current developments when applying motivational theory to OA publishing and potentially extending the existing frameworks.

The workshops confirmed the literature about different stakeholder views on OA publishing: researchers, the institutions or universities (who provide the publishing policy), and funding organizations have different attitudes and approaches to OA publishing [17,32], for instance, when it comes to their perception of re-usage quality, indicating a potential for further research to investigate such complexities. Results further show key differences regarding the assessment of publishing strategies and OA between the stakeholder groups. Whilst practitioners and academics saw OA as a way to increase the visibility of their paper, they highlight the role of journal ranking and impact factors as important criteria. While academics focused on reputation as the main criteria, they were also eager to spread their ideas and be known [35], which is quicker and more possible through OA publishing. Policy-makers took on a rather critical stance regarding the impact factor, noting that they are always referred to the same papers and that OA does not always lead to the additional value claimed [37].

The survey, developed on the results gained from the workshops, reconfirmed that institutional context plays a role, but other factors impact the target group more. Participants strongly reflected OA publishing as a highly individualist matter that is embedded within an institutional publishing culture that continues to strongly emphasize reputation and rankings. OA, or publications free of charge are seen as a good opportunity for students, but even when costs are not an issue, the motivation to publish OA is small when there is no institutional support or the institution does not have an OA strategy. When personal strategies play a role, values are important, such as knowledge sharing, co-creation, knowing the rules of the game, and also avoiding the rat-race.

Emphasized here are the roles of indexes and visibility, and in this context digital literacy becomes important. Whilst OA is understood as everything and anything ranging from being online to open source, the majority of users were unfamiliar with more nuanced, but important, issues such as Plan S and altmetrics whilst still claiming to know enough about OA to take important decisions regarding academic publications and career paths. This suggests that some aspects of OA publishing might be misunderstood, such as the options and different versions of OA publishing, even within our ICT-oriented and OA-interested target group. In line with previous results that investigated the perception of OA articles as of low research quality [15a], the results within our community also suggest that disciplines play an important role for the perception of research quality in OA journals. However, free publishing was not per se associated with lower quality, but OA publishing culture was contrasted to more prestigious journals or well-known publication venues. However, our academic community was aware of existing stereotypes that are associated with OA and research quality, as became apparent in the workshops. In the survey, users rated the quality of OA as very good, good, or fair, and 40.8% agreed that JeDEM publishes articles of high quality, even though some users perceived the paper quality as mixed. This suggests that, based on our interdisciplinary and case, not only disciplines but also cultures and the affinity of a research community to topics such as OA (or potentially other areas dealing with technology or openness) play an important role in the perception of research quality. It further suggests that the dominant (mis)perceptions regarding paper quality and OA that have been pointed out by previous research [43] (p. 729) do not have to persist in all contexts and communities. Moreover, if we assume a potential for a more nuanced understanding of OA aspects by most scholars, suggestive survey questions that tackle the issue of quality might be problematic. At the same time, we cannot see the opinion of our community, who clearly evaluated OA quality more positively, as a proof of a general opinion shift within academia. We thus suggest further investigations of the correlations among knowledge about OA, digital literacy, and the rating of quality.

Authors might benefit greatly from learning more about the detailed aspects of OA publishing, new metrics, and ways that help them to disseminate their work and achieve acknowledgement in new

digital contexts. For editors of OA journals, there is a clear need to understand authors' needs and fears, as well as to know about the factors of authors' publication choice. There is a need to provide support systems for authors and communities in order address their needs and by integrating tailored education and specific skill development. In addition, editors must also find ways of reaching and engaging with knowing users and informed communities, to provide authors more information about the digital impact and transformation of scholarly communication as well as to support their digital literacy education, which is often not a formal "qualification" but gained on the fly and through experience. The results, such as those gained about JeDEM and the institutional policy, show that users do not always know (or they misinterpret or misrelate) certain OA options, which might be due to stereotypical or incomplete information about OA. This could be one explanation for the gap between perceptions about OA and the actual practice that has been uncovered by previous research. We further identified a self-reported lack of digital literacy and knowledge when it comes to digital publishing. However, a detailed understanding of the current issues of open scholarship is necessary in order to better identify the unwanted effects or inappropriate applications of the metric systems in place, for instance, an artificial enhancement of individual or institutional reputation, index manipulation, and other emerging practices [60]. Editors could potentially play a role in connecting users to their communities and support critical stances and activism toward ethical and open scholarship, expanding the potentials for academic community engagement, and creating awareness for questionable practices that disrupt the role of communities or even violate trust in academic institutions and research output [60].

Institutions and funding agencies are advised not only to continue the development of OA strategies and provide support, but they should also encourage OA, digital literacy, and critical perspectives to be part of a curriculum of open scholarly publication and communication that is continuously changing and developing. Such institutional support for OA publication, thus, needs to consider the digital skills within the institution, and also the individual motivations and values authors may have, as well as the recognized dimensions of scholarly communication and publication such as indexing, visibility, impact, and reputation—in the context of an increasingly digital society with changing systems to measure these dimensions. One scientific challenge in this context will be to measure the indicators of value and credibility of different ways of scholarly output [71]. Authors further need to be supported in developing the skills for novel ways of showcasing their work, besides traditional communication or dissemination.

## 6. Conclusions

Research institutions, higher education institutions, and funding agencies are increasingly developing requirements to reflect the criteria of the funding practices. This means that the funding of research can be linked to requirements such as published findings and results in in OA journals or repositories, making publicly funded research accessible to a wider community. The research conducted here shows that different user groups have different views on OA publishing, but that some stereotypes and different understandings as to what OA publishing is or is related to (and what dimensions are important for successful publishing strategies) remain. Whilst personal strategies are important as they reflect authors' motivations and values, they are inevitably also linked to the institution. In addition to previous research pointing out the importance of disciplines [26–28,40], our case suggests that research communities and their affinity to thematic clusters such as digital openness or other digital topics are important for the perception of OA elements, while intrinsic factors such as academic reputation and recognition determine whether publications can be made accessible [50]. Citations and credit systems were also seen as being particularly important for visibility by our respondents. As recognition will likely be more connected to metric systems in the future, scholarly-led OA publishing will benefit from the integration of credit systems for different roles and stakeholders (authors and reviewers), which could be integrated into career plans that consider OA in the future.

There are two limitations in the studies conducted here. First, the workshops were held in 2016 and informed the survey that was run 3 years later. There is a need to communicate more regularly with users and to create more continuous data in the future. Secondly, the response rate of the survey was very low. On the other hand, the observations made in 2016 were reflected in the outcomes gained in 2019. This shows that in some respects OA, although several advances in terms of digitalization, specific digital tools, alternative metrics, and policy have been made, is still a large unknown for many authors. There is a clear need for enhancing a digital literacy that acknowledges more nuanced aspects of OA publishing. Whilst the benefits and detailed knowledge of OA publishing are clearly found within the closer OA community and more engaged individuals, it needs to go beyond this. There needs to be a stronger institutional impetus not only to develop the policy but also to provide the means and skills to ensure that academic staff can learn about digital literacy in the context of OA, and overcome barriers and stereotypes associated with it. At the same time, investigating the role and potential of external communities or NGOs for the motivation of publishing OA and support of literacies will be important. In line with previous studies on publication choice and motivational factors, our case study in an interdisciplinary context of OA-interested individuals confirmed the importance of impact factors, reputation, and the influence of institutional context for those already interested in OA publishing. Further research should undertake a more detailed investigation of the hurdles and enablers of a more nuanced understanding of the subject. The low response rate was due to a problem with the journal system itself that was identified only after the first announcement was made on the journal platform. The old platform enabled editors to send personalized emails to all the registered users. This functionality was not supported by the new OJS system, so only a "generic" notification could be sent. Thus, the second announcement was supported by an email sent out using one of the author's academic email account. PKP (Public Knowledge Project), the developer of the platform was informed of this problem, and it is an issue to be addressed in a release of the platform in 2021. The authors aim to send out the survey again in order to regularly gain insights into the journal's user community. By understanding how authors choose where and how they publish, editors can find a way of reaching, engaging, and knowing users as well as support them in gaining more knowledge about the digital transformation of scholarly communication. The mixed-method approach showed that OA consists of complex dimensions when it comes to authors' publishing motivations, and requires corresponding digital literacies. The results gained point out the need for additional conceptualizations of publishing strategies and digital literacies. The next steps are therefore to gain further insights in order to reduce misconceptions and stereotypes regarding OA, to help the development of the authors' university OA policy in the context of the digital transformation of education, and the development of a digital literacy plan for staff, students, and the wider researcher community.

**Author Contributions:** N.E. and J.S. contributed equally to this manuscript. All authors have read and agreed to the published version of the manuscript.

**Funding:** This research received no external funding.

**Acknowledgments:** Thanks to Thomas Pfeffer, Department for Continuing Education Research and Educational Technologies, Danube University Krems, Austria, for reading the manuscript and providing us with valuable feedback.

**Conflicts of Interest:** The authors declare no conflict of interests.

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
