# Peer review of "Open Access Perceptions, Strategies, and Digital Literacies: A Case Study of a Scholarly-Led Journal"

_publications, doi:10.3390/publications8030044_

Round 1

Reviewer 1 Report

The paper is interesting but should be improved.

Introduction

The goal of the paper should be clearly defined (there are different descriptions, cf. lines 116, 385 and 525), perhaps with a narrower focus on one or two variables (attitudes and literacy, for instance). The introduction should clearly state the societal and scientific challenges of the topic. An illustration (model of the explored dependent and independent variables) would be helpful.

Overview 

No comment; relevant literature is cited. Perhaps one part of the introduction could move to the literature overview.

Methodology

The section should explicitly state that the paper presents results of a re-analysis of data produced for operational purposes (line 385) and that this is not a general survey but a case study of a specific journal (JeDem) in a specific field (political science). The nature of the work should be mentioned in the title. This doesn't reduce the interest of the paper but clearly inform the reader what she/he can expect.

Results

The presentation is descriptive and partly narrative, and I can't see any relationships between, for instance, literacy, attitudes and/or behaviours, or between institutional support and attitudes. No statistical test of probability?

Discussion

The discussion of limitations (line 614) should be part of this section. Again, a model (illustration) would be helpful. Are these results relevant for other journals, other research fields, other countries? It would be helpful, too, to add three or four headings and to highlight main results. What is specific, different with the JeDEM journal and community?

Conflicting interest

The paper should clearly mention that both authors are managing editors of the journal which is object of the case study.

Author Response

Dear Reviewer,

thank you for your review and the comments. We have addressed these as follows:

The goal of the paper should be clearly defined (there are different descriptions, cf. lines 116, 385 and 525), perhaps with a narrower focus on one or two variables (attitudes and literacy, for instance).

We c made the goals clearer of the research and paper clearer (lines 13 f., 116, 170-177), pointing it out for example in the introduction and in general when referring to factors. In addition, we reduced the focus of the factors, addressing only attitudes/personal strategies and literacies.

The introduction should clearly state the societal and scientific challenges of the topic.     

We addressed these challenges throughout the whole paper, but specifically in introduction  (approx. line 79),  and in the outlook.

An illustration (model of the explored dependent and independent variables) would be helpful.    

We added a paragraph about case study methodology and why chose this method. Methodologically, though we follow a different approach with no exploration of such variables as there would be in a classic survey analysis.

Overview: No comment; relevant literature is cited. Perhaps one part of the introduction could move to the literature overview.           

We moved shortened the introduction and moved most of the literature in the following section, the literature overview. The literature was also further condensed.

Methodology: The section should explicitly state that the paper presents results of a re-analysis of data produced for operational purposes (line 385) and that this is not a general survey but a case study of a specific journal (JeDem) in a specific field (political science). The nature of the work should be mentioned in the title. This doesn't reduce the interest of the paper but clearly inform the reader what she/he can expect.      

We mentioned that it is a re-analysis of data and embedded it in the section on the methodology. In addition, we made it clearer that it is a case study, pointing this out in the title and adding a description as a section in its own right in the section on the research design. We also repeated these two aspects throughout the paper (e.g. also in the abstract and introduction).

Results: The presentation is descriptive and partly narrative, and I can't see any relationships between, for instance, literacy, attitudes and/or behaviours, or between institutional support and attitudes. No statistical test of probability?             

We did not add statistical tests as this was not part of the paper.    This was not the aim of the paper, so we made the aims of the reserach clearer in order to avoid such confusion.

 Discussion: The discussion of limitations (line 614) should be part of this section. Again, a model (illustration) would be helpful. Are these results relevant for other journals, other research fields, other countries? It would be helpful, too, to add three or four headings and to highlight main results. What is specific, different with the JeDEM journal and community?    

Yes, we think the results are relevant for researchers, and editors too (especially scholarly led ones who do not have the specific training) – so we have not only strengthened this issue, we indicate it in the introduction and the outlook. We also considered   What is different with our community and case – this has been addressed in more details and the added value of our case has been pointed out (see 3.2.)

We have also moved the description of the limitations to the discussion part as suggested.

Conflicting interest: the paper should clearly mention that both authors are managing editors of the journal which is object of the case study.

We would not see our role as a conflict of interest but agree that it should be made clear in the paper. We have addressed this a few times throughout the whole paper, e.g. in  the abstract  and in the introduction.

Reviewer 2 Report

line 39: What is this about? [10][10][10][10][10][10]

line 90-95 is only repetition; why?

The Introduction is very repetitive and long.

Author Response

Dear Reviewer,

thank you for your comments, we addressed them as follows:

line 39: What is this about? [10][10][10][10][10][10]

This was an error/bug in the Endnote file and have resolved it.

line 90-95 is only repetition; why?

We deleted the repetition here as well as one that occurred later in the introduction!

The Introduction is very repetitive and long.

We shortened the introduction to focus on the overview of the topic and the research at hand. Most of the literature it contained was moved to the following section and then condensed.

Reviewer 3 Report

It’s a well-structured and written paper but, in my opinion, the manuscript has three remarkable deficiencies and limitations:

1) Review
The bibliographic review is too general, too basic. These sentences are not true:
“To date, little is known about how authors select a publication venue or repository, particularly in OA publications”, or “There are only few studies that look at scholars’ attitudes regarding publishing”
There are a lot of surveys and studies about open access publication and OA perception that should be reviewed. For example:

Taylor and Francis perform an annual survey about open access (since 2013)
- 2019: https://authorservices.taylorandfrancis.com/researcher-survey-2019/
- 2013: https://www.tandf.co.uk//journals/explore/Open-Access-Survey-March2013.pdf

European University Association has also several surveys:
- EUA Open Access Survey Results: 2017-2018.
- Open Access in European universities: Results from the 2016/2017 EUA institutional survey
- Open Access in European universities: Results from the 2015/2016 EUA institutional survey

European Comission (2012). Survey on open access in FP7 https://ec.europa.eu/research/science-society/document_library/pdf_06/survey-on-open-access-in-fp7_en.pdf
European Union (2012). Online survey on scientific information in the digital age. 10.2777/7549

Several authors:
- Nicholas, David et al (2017). Early career researchers and their publishing and authorship practices. Learned Publishing 30 (3), 205-217. (This author has a lot of similar studies)
- Serrano-Vicente, R. et al (2016). “Open Access Awareness and Perceptions in an Institutional Landscape”. Journal of Academic Librarianship. http://dx.doi.org/10.1016/j.acalib.2016.07.002 (You can find a wide bibliographical review)

2) Methodology
The case of study (readers of JEDEM) is quite limited in comparison with the population analysed by most of similar studies that have been made. Moreover, the authors have obtained a low response rate in the survey, and there is a disconnection in topic and in time between the two phases of the methodology (the workshop and the survey).

3) Results and discussion
There is a scarce and poor discussion with some of the multiple studies published previously on this topic. The text doesn’t make comparison with previous surveys, it has no dialogue with them.

For these reasons, I have serious doubts about the interest and the real contribution of this paper to “Publications” community.

Author Response

Dear Reviewer,

thank you for your comments, we have addressed them as follows:

(x) English language and style are fine/minor spell check required   

 The manuscript has been proofread by an external proofreader after we made our revisions.

The bibliographic review is too general, too basic. These sentences are not true: “To date, little is known about how authors select a publication venue or repository, particularly in OA publications”, or “There are only few studies that look at scholars’ attitudes regarding publishing” There are a lot of surveys and studies about open access publication and OA perception that should be reviewed. Recommends several reports and authors, f.i. Nichols, Serrano-Vicene etc. 

Thank you for pointing this out. We have rephrased the generic formulations. In addition, we considered and discussed the literature recommended, we have included several of the texts, for example:

*Taylor & Frances 2019: https://authorservices.taylorandfrancis.com/researcher-survey-2019/ (line 98)

*European Union (2012). Online survey on scientific information in the digital age.

*European Commission (2012). Survey on open access in FP7 https://ec.europa.eu/research/science-society/document_library/pdf_06/survey-on-open-access-in-fp7_en.pdf

(line 98-105).

*EUA Open Access Survey Results: 2017-2018.

Further included in this research are

*Nicholas, David et al (2017). Early career researchers and their publishing and authorship practices. Learned Publishing 30 (3), 205-217) for similar studies

(included from 110 etc., as well as other scholars, f.i. Kim, 2007.)

*Serrano-Vicente, R. et al (2016). “Open Access Awareness and Perceptions in an Institutional Landscape”. Journal of Academic Librarianship (wide bibliographical review)

Included from line 127, as well as Tenopir et al.

(line 133).

Methodology: The case of study (readers of JEDEM) is quite limited in comparison with the population analysed by most of similar studies that have been made. Moreover, the authors have obtained a low response rate in the survey, and there is a disconnection in topic and in time between the two phases of the methodology (the workshop and the survey). 

We referenced the other studies and surveys, which often (apart from the big publisher surveys) are for a specific segment (like one university, one segment like early career researchers), and point out the added value of our case as an interdisciplinary target group that extends to different universities (and other factors). We address in the the abstract our research as a process of evaluation to make this clearer. In addtiion, we also point out the need to communicate more regularly with the users (and in conclusion).

We also dded a section about case study in the research design section and the discussion (para 2)

There is a scarce and poor discussion with some of the multiple studies published previously on this topic. The text doesn’t make comparison with previous surveys, it has no dialogue with them.            

In the text we have pointed out some possible connections to previous research in the conclusion, for instance, the gap between general perception and the actual practice of OA publishing. In the discussion and conclusion we have summarized the insights gained from our case.

Round 2

Reviewer 1 Report

The author response addresses the main issues, and I think that the revised version can be published. 

Author Response

Thank you for your comments

Reviewer 3 Report

The authors have addressed some of my comments, but I still want to remark an inaccurate use of “gold” and the need compare the results obtained with previous studies.  

1 Meaning of “gold”

There is an inaccurate use of “gold” in the manuscript when the term is applied to “journal”. We can say “gold OA” (as the opposite to the green OA, the repositories) but when we say “gold open access journal” we refer to journals funded by APC. In the case of public financing of the journals, it’s very common to refer them as “diamond or platinum” journals.

For this, it’s necessary to check this term in the whole text to adjust the meaning.

- Lines 14, 93, 492

Inaccurate use of “gold open access journal”. JEDEM is an “open access journal” but not a “gold open access journal”

- Lines 141-142

Inexact use of “gold open access”:

- “The golden approach provides the public with free access but imposes publication fees of several hundred up to several thousand Euros per article on its authors”.

The gold approach means the use of journals whether they are funded by APC (authors) or by public funds (universities, research centres, etc.). According to DOAJ, only 27 % of the journals charge APC. Therefore is not exact to say that “the golden approach … imposes publications fees .. on its authors”

- Line 502

What does mean “fair” in this sentence: “their institution supported publication in gold and fair OA journals”?

Which is a “fair” OA journal? Are you sure that the sentence has been understood by the people surveyed?

2 Discussion

There are no changes in this section in order to compare your results with previous ones, as I mentioned in my first report. It is important to make comparison with previous surveys and stablish dialogue with them.

3 Other comments

- Lines 209-210

“Repeated surveys have been conducted for example by publishers such as Elsevier in 2019 and Springer in 2020 in order to access author perceptions through their registered users.”

Reference of these studies should be provided.

- Another reference, if you want to enhace your list of studies about OA perception:

Ruiz-Pérez, Sergio; Delgado-López-Cózar, Emilio (2017). “Spanish researchers’ opinions, attitudes and practices towards open access publishing”. El profesional de la información, v. 26, n. 4, pp. 722-734. https://doi.org/10.3145/epi.2017.jul.16

Author Response

Dear Reviewer,

thank you very much for re-reading our manuscript so carefully and pointing out the weaknesses and errors - we appreciated this careful reading very much! We have considered all the comments made and hope to have taken out the errors and strengthened our arguments.

Inaccurate use/meaning of gold: for this, it’s necessary to check this term in the whole text to adjust the meaning.

  • We went through the whole text checking this, as it is correct that "gold" OA does not mean without APC, so we now describe the journal in this case as "diamond/platinum" throughout the paper.
  • Consequently, for the diamond definition, we quoted: Öchsner, A., "Publishing Companies, Publishing Fees, and Open Access Journals," in Introduction to Scientific Publishing: Backgrounds, Concepts, StrategiesBerlin, Heidelberg: Springer Berlin Heidelberg, 2013, pp. 23-29 (ref. nr. 8) .There, on page 28, the author defines it as platinum: “There are also open access journals which do not charge any fees at all to the author since the costs are covered by institutional subsidies and/or voluntary work and this concept is sometimes called platinum open access.” We included this reference and definition (line 38-39), then in line 40-41 we distinguish between gold, diamond/platinum, drawing on Öchsner.

Further corrections based on the inaccurate use of "gold":

  • In line 14, “gold” was removed as the abstract should not contain the definition. 
  • In line 54 it was removed, and details regarding the type of OA that OA working groups e,g. in Austria have been included instead.
  • It was also removed from line 98 as the focus should here is on the journal’s interdisciplinarity.
  • Line 144: the definition of gold was clarified, also included the reference by Singh, S. and Morrison, H. (2019, 27.08.2020). OA journals non-charging and charging central trends 2010 – 2019 (ref nr. 32) regarding the amount of journals charging APCs.
  • We added further details about types of OA in lines 145-149.
  • We pointed out that the journal investigated is diamond/platinum on line 350.

2. Use of the word "fair": Which is a “fair” OA journal? Are you sure that the sentence has been understood by the people surveyed?

  • Our suggestion is that by fair we mainly mean fair to authors and as researcher-centric, and included a reference to the FOAA, the Fair Open Access Alliance (reference 78) and an article about it (Rooryck, 2016, ref 79)
  • We addressed the definition on line 530.
  • The criticism that users might have lacked a reference of gold or fair open access is valid, in particular since these terms can often lead to confusion even for researchers familiar with OA. In future, surveys we will cdevelop should provide more context or a link.

3. Discussion: There are no changes in this section in order to compare your results with previous ones, as I mentioned in my first report. It is important to make comparison with previous surveys and establish dialogue with them.

  • We tried to compare our results with the previous surveys where possible (in the discussion but also the conclusion, and the results).
  • Comparison has been made for instance regarding perceptions of OA regarding motivational factors and the perception of research quality. (See line 673 ff for perception of quality, and 734, 519, 601, 622 etc.)
  • One point we would like to mention is that some surveys might not be directly comparable, as different cohorts and universities have been the focus of the previous studies, this has also been pointed out in the literature review (line 224). However, we certainly can compare attitudes about OA publishing and we decided to emphasise motivations and quality.  

“Repeated surveys have been conducted for example by publishers such as Elsevier in 2019 and Springer in 2020 in order to access author perceptions through their registered users.” Reference of these studies should be provided

  • We included the following references (the first two are emails to one of the authors, lines 227-231) -

    Springer Nature, "Tell us about how you decide which journal to submit to," N. Edelmann, Ed., email ed, 2020.[ref nr. 46]

    Springer Nature, "The State of Open Data 2020: Survey Now Open," N. Edelmann, Ed., email ed, 2020.[ref. nr. 47]

    Taylor & Francis, "Taylor & Francis researcher survey 2019," Taylor & Francis Group 2019, [ref. nr. 48]

Another reference, if you want to enhace your list of studies about OA perception: Ruiz-Pérez, Sergio; Delgado-López-Cózar, Emilio (2017). “Spanish researchers’ opinions, attitudes and practices towards open access publishing”. El profesional de la información, v. 26, n. 4, pp. 722-734. https://doi.org/10.3145/epi.2017.jul.16

  • Thank you for the reference - we read the article and found it useful, so we have included it (lines 198-205, and 688-689, ref. nr. 45.)

Once again, thank you for your review!

Best regards

The Authors